# The Dominance of Pretransitional Effects in Liquid Crystal-Based Nanocolloids: Nematogenic 4-methoxybenzylidene-4′–butylaniline with Transverse Permanent Dipole Moment and BaTiO_3_ Nanoparticles

**DOI:** 10.3390/nano14080655

**Published:** 2024-04-09

**Authors:** Aleksandra Drozd-Rzoska, Joanna Łoś, Sylwester J. Rzoska

**Affiliations:** Institute of High Pressure Physics, Polish Academy of Sciences, ul. Sokołowska 29/37, 01-142 Warsaw, Poland; arzoska@unipress.waw.pl (A.D.-R.); joalos@unipress.waw.pl (J.Ł.)

**Keywords:** liquid crystals, broadband dielectric spectroscopy, nanocolloids, critical phenomena, phase transitions, dynamics

## Abstract

The report presents static, low-frequency, and dynamic dielectric properties in the isotropic liquid, nematic, and solid phases of MBBA and related nanocolloids with paraelectric BaTiO_3_ nanoparticles (spherical, *d* = 50 nm). MBBA (4-methoxybenzylidene-4′–butylaniline) is a liquid crystalline compound with a permanent dipole moment transverse to the long molecular axis. The distortions-sensitive analysis of the dielectric constant revealed its hidden pretransitional anomaly, strongly influenced by the addition of nanoparticles. The evolution of the dielectric constant in the nematic phase shows the split into two regions, with the crossover coinciding with the standard melting temperature. The ‘universal’ exponential-type behavior of the low-frequency contribution to the real part of the dielectric permittivity is found. The critical-like pretransitional behavior in the solid phase is also evidenced. This is explained by linking the Lipovsky model to the Mossotti catastrophe concept under quasi-negative pressure conditions. The explicit preference for the ‘critical-like’ evolution of the apparent activation enthalpy is worth stressing for dynamics. Finally, the long-range, ‘critical-like’ behavior of the dissipation factor (*D* = *tgδ*), covering the isotropic liquid and nematic phases, is shown.

## 1. Introduction

Liquid crystalline (LC) mesophases can exist between the isotropic liquid and the solid phase of rod-like molecular materials [1,2,3,4,5,6]. In the simplest case of the isotropic liquid (I)–nematic (N) transition, ‘freezing’ of solely orientational, uniaxial ordering occurs. It is associated with long-range, critical-like pretransitional effects despite the (weakly) discontinuous character of the I–N transition. They result from prenematic fluctuations, whose correlation length ξ and lifetime $\tau_{fl.}$ follow the critical-like behavior [1,2,3,4,5,6]:(1a)$$\tau_{fluct.}\left(T\right)=\tau_{0}{\left(T-T^{*}\right)}^{\nu =-1/2},$$
(1b)$$\xi \left(T\right)=\xi_{0}{\left(T-T^{*}\right)}^{\phi =-1},$$
where the subscript ‘*fluct*.’ stresses the link to multimolecular pretransitional fluctuations; $\tau_{0}$ and $\xi_{0}$ are prefactors; $T>T_{I–N}$ and $T^{*}=T_{I–N}-{\Delta T}^{*}$; $T^{*}$ is the hypothetical continuous phase transition temperature hidden in the nematic phase; ${\Delta T}^{*}$ is the phase transition discontinuity metric; and $T_{I–N}$ is the isotropic–nematic phase transition temperature.

Generally, the transition from the isotropic liquid to different LC mesophases can be considered a symmetry-limited freezing/melting transition. The temperatures of such phase transitions are traditionally referred to as the clearing temperature: in the given case, $T^{C}=T_{I–N}$ [1,2,3,4,5,6]. Rod-like LC materials are strongly susceptible to endogenic and exogenic impacts. The impact of pressure [5] and the electric field [1,2,3,4,5,6] are particularly important. The latter, essential for applications, stresses the significance of broadband dielectric spectroscopy (BDS) studies [3,5,6].

For the endogenic impacts, molecular admixtures [7] or solid-state nanoparticle (NP) additions are notable [8,9,10,11,12,13,14,15,16,17,18,19,20,21,22,23,24]. Their influence is so significant that a new category of LC + NP nanocolloids and nanocomposites was established [8,9,10]. The unique fundamental properties and expectations for innovative applications reinforced this topic. It can be seen from the fact that the term “nematic and nanoparticles” appeared in 135 papers in the year 2000, and in 2920 reports in 2023 [25].

For LC + NP nanocolloids, studies focused on the impact of pretransitional fluctuations remain surprisingly limited, although their significance for ‘pure’ LC materials is well evidenced [1,2,3,4,5,6]. To the best of the authors’ knowledge, the first report addressing explicitly this issue was published in 2016 [26]. It discussed changes in the dielectric constant for dodecylcyanobiphenyl (12CB) and paraelectric BaTiO_3_ nanoparticle nanocolloids. 12CB is the LC material with the Solid (S)–Smectic A (SmA)–Isotropic Liquid (I) mesomorphism. The dominant impact of fluctuations was observed in pure LC compounds and related nanocolloids. It manifested via the following pretransitional effects [26]:(2)$$\varepsilon \left(T\right)=\varepsilon^{*}+a\left(T-T^{*}\right)+B{\left(T-T^{*}\right)}^{\alpha =1/2}\;\mathrm{for}\;T>T^{C},$$
(3)$$\varepsilon \left(T\right)=\varepsilon^{**}+a^{\prime }\left(T^{**}-T\right)+B^{\prime }{\left(T^{**}-T\right)}^{\alpha =1/2}\;\mathrm{for}\;T<T^{C},$$
where Equation (2) is for the isotropic liquid phase and Equation (3) is for the SmA mesophase. $T^{**}$ is the temperature of a hypothetical continuous phase transition extrapolated from SmA mesophase.

Subsequent research reports showed the analogous pretransitional anomalies in nanocolloids based on rod-like LC with solid–nematic–isotropic mesomorphism (in pentylcyanobiphenyl (5CB) + NPs) or solid (S)–smectic A (SmA)–nematic (N)–isotropic (I), in undecylcyanobiphenyl (11CB + NPs), and octyloxycyanobiphenyl (8OCB + NPs) [27,28,29,30,31,32,33].

In refs. [34,35], the pretransitional anomaly described by Equation (2) was explained by the cancellation of the contribution from permanent dipole moments to dielectric permittivity within prenematic fluctuations in the isotropic liquid surrounding. It results from the basic feature of the nematic phase, namely, the equivalence of $\vec{n}$ and $-\vec{n}$ directors indicated the averaged local uniaxial arrangement of rod-like molecules. The cancellation can occur for rod-like molecules with the permanent dipole parallel to the long molecular axis, as for 12CB, 11CB, 5CB, or 8OCB. When the clearing temperature is approached, the volume occupied by fluctuations increases according to Equation (1) $V_{fluct.}\sim \xi^{3}\propto {\left(T-T^{*}\right)}^{-3/2}$. Consequently, the dielectric constant within fluctuations is qualitatively lower than for the ‘non-ordered’ surroundings: $\varepsilon_{fluct}.<<\varepsilon_{surronding}$. On approaching the I–N transition, the contribution from the volume related to fluctuations finally dominates, and the following crossover occurs [27,28,29,30,31,32,33,34,35]:(4)$$d\varepsilon /dT>0\leftarrow d\varepsilon /dT=0\leftarrow d\varepsilon /dT<0\;\mathrm{for}\;T_{I–N}\leftarrow T,$$
in agreement with Equation (2).

The addition of nanoparticles (paraelectric BaTiO_3_, C_60_ fullerenes) preserved the form of Equation (2) with the exponent α=1/2. The mentioned mechanism should be absent for rod-like molecules with the transverse (perpendicular) dipole moment with respect to the long molecular axis. The classic example is MBBA (4-methoxybenzylidene–4′–butylaniline), for which linear changes of dielectric constant are observed in the isotropic liquid on approaching the I–N transition [3,6,35,36,37,38,39,40,41,42].

The strongly discontinuous fluid LC mesophase–solid transition has hardly been studied, both in pure LC materials and related nanocolloids. This issue recalls the general challenge of the melting/freezing transition between liquid and solid crystal transition. For the canonic case, no pretransitional effects are expected here [43,44,45]. Nevertheless, a set of results shows weak and range-limited pretransitional effects in the solid phase, which is not amenable to reliable parameterization [43,44,45,46,47,48,49,50]. For this phenomenon, the transition from the liquid but partially ordered LC mesophase to the solid crystalline phase may be an important complementation, particularly when considering the supplementary impact of nanoparticles.

This report presents the results of high-resolution, broad temperature range BDS studies focused on pretransitional effects accompanying the isotropic–nematic (I–N) and nematic–solid (N–S) transitions in MBBA and its nanocolloids with paraelectric BaTiO_3_ nanoparticles. MBBA is the LC compound with the transverse orientation of the permanent dipole moment. So far, such studies have been carried out only for nanocolloids based on LC molecules with the parallel arrangement of dipole moment. The results presented also offer new results for low frequency-related dielectric properties, still constituting a cognitive challenge.

## 2. Materials and Methods

Studies were carried out using 4-methoxybenzylidene-4′–butylaniline (MBBA), purchased from Sigma-Aldrich, with the following reference mesomorphism [3,39]: Solid←294.3 K←Nematic←318.0 K←Isotropic. Its structure is associated with the permanent dipole moment μ=1.983 D directed at the angle β≈86° to the long molecular axis [51].

MBBA was the first stable LC compound with a nematic phase near room temperature, which was important for fundamental studies and innovative applications. In 1968, Heilmeir’s RCA group used MBBA for the first stable experimental liquid crystal display panel [52].

In the early 1970s, nematogenic pentylcyanobiphenyl (5CB) was synthesized [52]. It represents a large family of LC compounds (n-cyanobiphenyls nCB, n-oxylcyanobiphenyl nOCB) that are more stable and less hygroscopic than MBBA [3,52]. In these materials, the relatively large permanent dipole moment (μ~5D) is approximately parallel to the long molecular axis. Such properties are beneficial for applications and more convenient for fundamental studies than features of MBBA. In recent decades, MBBA has been particularly interesting for fundamental research as a classic example of a rod-like LC molecule with the transverse dipole moment [3,6,7,35,36,37,38,39,40,41,42,52].

Paraelectric BaTiO_3_ nanopowder (paraelectric, diameter d=50 nm) was purchased from US Research Nanomaterials, Inc.: see ref. [53] showing the link to the characterization of these nanoparticles. Mixtures of the liquid crystal and nanoparticles were sonicated at a temperature above the isotropic–nematic phase transition for 4 h to obtain homogeneous suspensions. Studies were carried out for nanocolloids with nanoparticle concentrations of 0.05%, 0.1%, 05%, and 1% weight (mass) fractions. Our experience shows that significant nanoparticle sedimentation can occur at concentrations above 1% in such systems [32,33]. It can be avoided by introducing a macromolecular surfactant [8,9,10], but it significantly distorts the dielectric response by introducing the molecular admixture, which can also lead to a shift in the clearing temperature ([7] and refs therein). The preparation of nanocolloids with such few nanoparticles is always a laboratory challenge. It requires a large amount of expensive (high-purity) liquid crystalline material.

The paraelectric properties and the globular form of the nanoparticles (NPs) allowed us to minimize their phase and shape-related orientational impacts on the host LC system. The authors’ experience shows that for the type of systems tested in the given report, sedimentation of nanoparticles can occur at concentrations above 1%. One can avoid this by introducing a macromolecular surface agent, but even a minimal molecular admixture can significantly shift the clearing temperature [7]. Moreover, the additional macromolecular agent can significantly distort the dielectric response, making a reliable analysis difficult. The stability of nanocolloidal samples was tested by measuring the dielectric constant and electric conductivity in a special capacitor with rectangular plates: 20 mm in length and 5 mm in width, with two sections. Tests were conducted in the isotropic liquid and the nematic phase to the ‘longitudinal’ and ‘transverse’ positions. For concentrations below x%=0.5%, no changes in dielectric properties were noted during the whole experiment. A small difference appeared (for concentration x%=1%) after >1 h of observation. This factor was removed by strong electric field mixing, using sine–wave packages (f=1 kHz) and lasting 10 ms, with voltage U=200 V. Worth recalling are also polarized microscopy observations in 5CB-based nanocolloids, which is a nematogenic compound with density and viscosity similar to MBBA. Tests were carried out in the ‘horizontal’ and ‘vertical’ arrangements, confirming the above conclusions [32,33].

Generally, the problem of stability and time-dependent sedimentation constitute a crucial issue in experimental studies of LC-based nanocolloids, which is also essential for possible applications. Notably, the behavior described in the given report should be considered for the given type of LC reference system and nanoparticles. The nanoparticles’ size, type, and topology can be significant for the ‘parasitic’ sedimentation. Notably, some nanoparticles can form ‘structured’ clusters within the nematic matrix. Such behavior was observed for Au nanoparticles in thin layers or LC mesophases [54]. However, it was a different case with regard to the size and type of nanoparticles discussed in the given report. The results presented are for bulk-sized samples; in this way, the size constraints are avoided. They are related to the orientational impact of capacitor plates, always significant for LC rod-like systems studied using micrometric distances between plates [3,4,5,6,8].

The tested samples were placed in a flat parallel capacitor (diameter 2r=15 mm) made of Invar. The distance between the plates was d=0.2 mm. The capacitor was located in the Quattro Novocontrol facility, which allowed temperature control at ±0.05 K. It was connected to the Alpha Novocontrol broadband dielectric spectroscopy (BDS) analyzer, allowing a permanent 6-digit resolution for the applied U=1V monitoring voltage. BDS frequency scans were performed for 228 temperatures in the temperature range from 133 K to 360 K. Figure 1, Figure 2 and Figure 3 show examples of spectra for the real part of the dielectric permittivity. Imaginary parts of dielectric permittivity are shown in Appendix A.

Figure 1 also shows the location of the static domain related to the dielectric constant $\varepsilon =\varepsilon^{\prime }\left(f\right)$, where a shift in frequency has a minimal impact on the value of $\varepsilon^{\prime }\left(f\right)$. At frequencies below the static domain, values of $\varepsilon^{\prime }\left(f\right)$ strongly increase, it is the low-frequency (LF) domain. The relaxation ranges appear via distortions from the ‘near-horizontal’ static domain behavior for frequencies above the static range.

For the reference dielectric constant estimations, the frequency in the middle of the static domain was taken to take into account shifts associated with the broad range of tested temperatures. Noteworthy are spectra in the solid phase, which are significantly different from those in the nematic and isotropic liquid phases; they are presented in Figure 2 and Figure 3. The difference between spectra detected close and remote from the N–S transition is notable. In Figure 2, the schematic structure of MBBA is additionally shown.

## 3. Results and Discussion

Properties of the nematic phase of LC compounds are often tested for oriented samples, which is the natural consequence of the rod-like molecular structure [1,2,3,4,5,6]. It can be realized using the steady, strong electric field interacting with the permanent dipole moment coupled to LC molecules. However, it can yield only one direction of orientation of the long molecular axis. To obtain both ‘parallel’ and ‘perpendicular’ arrangements, a strong external magnetic field with a strength well above 1 Tesla is required [1,2,3,4,5,6]. The alternative method constitutes the special treatment of capacitor plates, which are covered by a polymeric layer supporting the preferred orientation of rod-like molecules. However, it is possible only for narrow gaps between capacitor plates, usually 1–10 μm [2,3,4,5,6,8,9,10]. Notably, the standard voltage for dielectric studies U=1 V yields the intensity of the extreme electric field $E\sim 10^{-6}÷10^{-5}\;V/m$, i.e., in the range of the nonlinear dielectric spectroscopy [7]. For the gap between plates used in this report, the intensity is qualitatively lower, namely $E=0.2\times 10^{-3}\;V/m$. Additionally, separation of the contribution from the thin polymeric layer covering the capacitor plates is a difficult task.

Recent studies of nCB and nOCB liquid crystalline compounds showed that adding BaTiO_3_ nanoparticles (d=50 nm) can create a unique endogenic permanent orientation of rod-like LC molecules in bulk [30,31,32,33]. It is reached without any strong external field (electric, magnetic). This report follows this path by focusing on tests in a bulk LC compound subjected solely to the impact of nanoparticles in an LC nematogenic compound with a transverse location of the permanent dipole moment.

Temperature changes in the dielectric constant $\varepsilon \left(T\right)$ for MBBA and MBBA + NP nanocolloids, extending over the isotropic liquid, nematic, and solid phases are presented in Figure 4.

The supplementary distortions-sensitive $d\varepsilon \left(T\right)/dT$ insight validates the existence of pretransitional effects, as shown in Figure 5. Figure 4 and Figure 5 reveal the significant influence of nanoparticles on phase transition temperatures: the clearing temperature ($T^{C}=T_{I–N}$) and the solidification temperature (N–S transition). Their values are also presented in Table 1.

Worth stressing is the significant shift in $T^{C}$ when changing the amount of NPs. For nanocolloids tested earlier based on 5CB, 11CB, 12CB, and 8OCB LC compounds with the addition of the same nanoparticles, the shift in $T^{C}$ was negligible [26,27,28,29,30,31,32,33]. These compounds are associated with the permanent dipole moment parallel to the long molecular axis. For MBBA, the transverse/perpendicular location of the permanent dipole moment takes place. Hence, the question arises if this factor can be responsible for the difference in $T^{C}\left(x_{NP}\right)$ dependences.

In any case, there may be a difference in the interaction between the nanoparticles in question and the surrounding LC molecules with the ‘parallel’ and ‘transverse’ dipole moments. For essentially paraelectric BaTiO_3_ (d=50 nm) nanoparticles, the appearance of super-paraelectric elements on the surface was suggested [55,56]. This may lead to a preference for Coulombic interactions with permanent dipole moments coupled to LC molecules, creating different local arrangements of LC molecules with the parallel and transverse positions of the permanent dipole moment.

The results presented in Figure 4 and Figure 5 also reveal specific patterns of $\varepsilon \left(T\right)$ evolutions in isotropic liquid, nematic, and solid phases. They reflect the impact of pretransitional effects driven by multimolecular pretransitional fluctuations.

### 3.1. Dielectric Constant Changes in the Isotropic Liquid Phase

For the isotropic liquid phase, $\varepsilon \left(T\right)$, the changes shown in Figure 4 seem to follow the linear behavior for nanocolloids and pure MBBA. The latter agrees with existing evidence and can be explained by the transverse position of the permanent dipole moment, as discussed in the Introduction. However, the derivative-based and distortion analysis, whose results are presented in Figure 5, reveals the existence of ‘hidden’ pretransitional effects in the isotopic liquid phase. Figure 6 and Figure 7 present the focused insight for two ranges of nanoparticle concentrations. Notable are the fair portrayals of the relation resulting from Equation (2):(5)$$\frac{d\varepsilon \left(T\right)}{dT}=a+B{\left(1-\alpha \right)\left(T-T^{*}\right)}^{-\alpha }=a+C{\left(T-T^{*}\right)}^{-\alpha },$$
where a,C=const.

Values of parameters related to the parameterization recalling Equation (5), and presented graphically in Figure 6 and Figure 7, are given in Table 2. In Figure 6, the derivative analysis revealed hallmarks of the pretransitional behavior, extending even up to $\sim T^{C}+20\;K$ for pure MBBA. It significantly shrinks when adding nanoparticles. The detected $d\varepsilon \left(T\right)/dT$ pretransitional changes agree with the behavior obtained in the isotropic phase of LC compounds with the ‘parallel’ location of the permanent dipole moment with the crossover given by Equation (4). Hallmarks of the pretransitional behavior are revealed for the regions well above the crossover at $d\varepsilon \left(T\right)/dT=0$, which are hidden in the non-accessible region close to $T_{I–N}$ and $T^{*}$. The detection of the discussed pretransitional effect can be associated with the fact that the permanent dipole moment is positioned in MBBA at the angle 86°; hence, a tiny component parallel to the long molecular axis exists, and its impact can be ‘canceled’ due to the prenematic ordering discussed above. It can be even more prominent because the uniaxial arrangement within fluctuations cannot be perfect. Moreover, the MBBA molecule exhibits a structural discrepancy from the rod-like approximation. All these can support the ‘prenematic cancellation’ of the component of dipole moment parallel to the long molecular axis of MBBA, leading to the portrayal of dielectric constant changes by Equations (3) and (5).

Figure 7 shows the behavior in the isotropic liquid phase in nanocolloids with greater concentrations of nanoparticles. It also can be portrayed by Equation (5), with parameters given in Table 2, for temperature from $T_{I–N}$ to ${\sim T}_{I–N}+60\;K$. Nevertheless, pretransitional $\varepsilon \left(T\right)$ changes in Figure 6 and Figure 7 are essentially different. For pure MMBA and slightly doped with nanoparticle nanocolloids, shown in Figure 6, the behavior resembles the pattern observed for LC compounds with the ‘parallel’ arrangement of the permanent dipole moment.

In such a case, the extrapolation to the singular temperature yields: $d\left(T^{*}\leftarrow \;T\right)/dT\;\to \;\infty$, which is associated with the amplitude B>0 in Table 2. For nanocolloids with greater concentrations of nanoparticles, $d\left(T^{*}\leftarrow \;T\right)/dT\;\to -\infty$, which is associated with the amplitude *B* < 0 in Table 2. It is an anomalous pretransitional behavior that has not been observed so far. In the authors’ opinion, it can be considered the emergence of the ‘perpendicular’ arrangement of rod-like molecules with respect to the direction of the probing electric field within prenematic fluctuations. Prenematic fluctuations in the isotropic phase should show characteristic features of the subsequent nematic phase. In this context, qualitatively different anomalies presented in Figure 6 and Figure 7, respectively, should correlate with changes in ε(T) and dε/dT in the nematic phase immediately after passing $T_{I–N}$ on cooling.

### 3.2. Dielectric Constant Changes in the Nematic Phase

For pure MBBA and MBBA that is slightly doped with nanoparticles, a notable decrease in $\varepsilon \left(T\right)$ occurs when entering the nematic phase, as visible in Figure 4. A similar behavior was noted for the $\varepsilon_{//}$ component of dielectric permittivity in pure LC nematogens with the negative anisotropy of the dielectric constant Δε<0 coupled to the transverse dipole moment [3,4,39,57,58].

For MBBA-based nanocolloids with greater concentrations of nanoparticles, a notable increase in $\varepsilon \left(T\right)$ occurs when entering the nematic phase, as visible in Figure 4. A similar behavior was noted for the $\varepsilon_{⊥}$ component of dielectric permittivity in pure nematogens with the negative anisotropy of dielectric constant Δε<0, including MBBA [3,4,39,57,58].

Hence, there is a correlation between the behavior of the dielectric constant in the nematic phase just after I–N and the above suggestion of the explanation of ‘classic’ and ‘non-classic’ pretransitional effects in the isotropic phase, shown in Figure 6 and Figure 7.

On approaching the Isotropic–Solid transition in the nematic phase, a characteristic rise (pure MBBA and its slightly doped nanocolloids) or decrease (nanocolloids with greater concentrations of NPs) of dielectric constants occurs. Notably, $\varepsilon \left(T\right)$ changes remain approximately constant in this region, as shown by the horizontal lines in Figure 4 and Figure 5. For pure MBBA, the crossover temperature between domains close to I–N and N–S transitions coincides with the standard N–S melting temperature $T_{m}$. All these suggest changes in the arrangement of MBBA molecules when passing the hidden melting temperature, i.e., $T_{cross}\approx T_{m}$ and entering the hypothetical supercooled nematic domain.

When discussing the behavior of the dielectric constant in the isotropic and nematic phases in MBBA and its nanocolloids with paraelectric BaTiO_3_ nanoparticles, notable is the increase in the average value of the dielectric constant, reaching 21% for x=1% nanocolloid. A similar behavior was observed in 5CB, 8OCB, 11CB, and 12CB nanocolloids with paraelectric BaTiO_3_ nanoparticles [26,27,28,29,30,31,32,33]. All these suggest that paraelectric BaTiO_3_ nanoparticles can introduce an endogenic arrangement of dipole moment associated with molecules, leading to significant changes in dielectric constant, both in the nematic mesophase and isotropic liquid phase.

### 3.3. Dielectric Constant Changes in the Solid Phase

Generally, no pretransitional effects are expected for the canonic melting/freezing discontinuous transition between the liquid and solid phases [43,44,45]. Nevertheless, there is evidence of a weak and range-limited pre-melting effect on heating from the solid phase. The basic model explanation suggests the fragmentation of the solid into crystalline grains covered by quasi-liquid nanolayers [46,59,60,61,62,63]. However, the volume associated with these nano-layers is (very) minimal compared to the volume occupied by solid, crystalline grains. It can explain the mentioned weakness and parameterization problems for pretransitional effects in the solid state. Generally, monitoring using different physical methods simultaneously detects the response from dominated solid grains and liquid nano-layers. Hence, the impact of the latter is (almost) negligible.

Recently, the authors of this report noted a promising exception for monitoring using dielectric properties, for which the response from liquid dielectrics can be qualitatively larger than from a solid crystal. It led to finding relatively strong pretransitional effects in the solid side of the melting/freezing transition, well portrayed by the following equation [64,65]:(6)$$\chi \left(T\right)=\varepsilon \left(T\right)-1=\frac{A}{T_{m,f}^{*}-T}+{\left(a+bT\right)}_{bckg},$$
and consequently:(7)$$\frac{\varepsilon \left(T\right)}{dT}=\frac{A^{\prime }}{{\left(T_{m,f}^{*}-T\right)}^{2}}+a,$$
where $T<T_{m,f}$, $T_{m,f}^{*}$ is the extrapolated temperature, ‘hidden’ in the liquid/fluid phase, $A^{\prime },A,\;a,\;b=const,\;$‘*bckg*’ index denotes the background term associated with the behavior of a non-disturbed solid phase, and $\chi \left(T\right)$ denotes the dielectric susceptibility.

The ‘background effect’ is associated with solid grains for which non-singular and ‘weak’ temperature dependence associated with the slope parameter a<<1 can be expected. This coefficient has a negligible impact on fitting via Equation (7). Such analysis, associated with Equations (6) and (7), have been proven for dielectric studies in nitrobenzene and linseed oil near melting/freezing transition [64,65].

Figure 8 shows similar behavior for the solid–nematic transitions in pure MBBA and its nanocolloids with x=0.01% and x=0.05% of nanoparticles. Notable is the very strong rise, both in the magnitude and the range, of the pretransitional effect in the solid phase when the concentrations of nanoparticles are increased. Regarding the origins of Equations (7) and (8), the model analysis by Lipovsky [66,67,68] was recalled in [64,65]. He considered the mentioned fragmentation into solid grains concept and focused on quasi-liquid nano-layers between grains, predicting their critical-like compressibility with the singular temperature $T_{m}^{*}$ located very close to the bulk temperature $T_{m}$:(8)$$\chi_{T}\propto \frac{1}{T_{m}^{*}-T},$$
where $T_{m}^{*}$ is the singular, ‘critical’ temperature almost coincident with the bulk discontinuous melting phase transition temperature.

Generally, the compressibility $\chi_{T}$ and dielectric susceptibility/dielectric constant tested in the given report are of qualitatively different physical magnitudes. They can coincide only for unique systems where the basic definition within the Physics of Critical Phenomena [2] can be implemented. For instance, it occurs in the paraelectric phase, where order parameter changes (polarization) can be tested concerning the coupled field parameter (electric field in the given case). Such behavior is explained within the Clausius–Mossotti local field model, generally valid for systems composed of non-interacting and weakly interacting dipole moments, including their ‘immersion’ in a mean-field surrounding [69,70,71,72,73,74,75,76,77,78]. Experimental dielectric studies of liquids under nano-constraints showed that they can mimic the behavior under quasi-negative pressure conditions [77,78,79,80], i.e., the ‘rarefication’ (isotropic stretching), which has to weaken intermolecular and inter-dipolar interactions. Such nano-constrained conditions can be essential for quasi-liquid nano-layers in the solid-state pre-melting domain. Therefore, one expects a description within the Clausius–Mossotti local field model, leading directly to the so-called Mossotti Catastrophe associated with the following behavior of dielectric susceptibility [72,73,74,75,76,77]:(9)$$\chi \left(T\right)\propto \frac{1}{T-T_{C}},$$
where $T_{C}$ is the hypothetical ‘ferroelectric’ phase transition.

According to the authors, the combination of the Lipovsky model with the above considerations regarding the emergence of quasi-negative pressure, the validity of the Clausius Mossotti and Mossotti Catastrophe local field models, and the particular resolution of dielectric studies can explain the results shown in Figure 8 and described by Equations (6) and (7), as well as that in refs. [64,65].

When discussing the results of BDS monitoring in the solid phase, it is worth paying attention to the manifestation of three phase transitions (TS1, TS2, TS3) in Figure 4 and Figure 5. The addition of nanoparticles to MBBA has no noticeable effect on the TS1 temperature, it may lead to a shift of approx. 3 K to the TS2 temperature, and for the TS3 temperature, this shift can increase to 7 K. Note the manifestation of processes associated with S1, S2, and S3 transitions in the frequency spectra of the imaginary part of dielectric permittivity presented in Appendix A. The correlation with earlier studies focused on solid MBBA [81,82,83,84,85,86] is notable.

### 3.4. Dielectric Permittivity in the Low-Frequency Domain

The description of the low-frequency (LF) changes in the real part of dielectric permittivity, i.e., below the static domain, remains a challenge despite its significance in practical implementations [3,39,87,88,89,90,91,92,93,94,95,96]. Figure 9 shows the temperature evolution of $\varepsilon^{\prime }(f,T)$ in the isotropic and nematic phases of MBBA and its nanocolloids with BaTiO_3_ nanoparticles for selected frequencies in the LF domain. There is a very strong increase in $\varepsilon^{\prime }\left(f\right)$ when lowering the monitoring frequency below the static reference adopted for *f =* 126 kHz in the given case. The average value of $\varepsilon^{\prime }\left(f\right)$ also increases the addition of nanoparticles.

The low-frequency rise in $\varepsilon^{\prime }\left(f\right)$ is typical for liquid dielectric materials, most often heuristically explained as the results of free ionic species, in LC materials often recalled as ‘ionic impurities’ [3,39,72,73].

In ref. [87], the behavior of $\varepsilon^{\prime }(T)$ focused on the low-frequency domain was tested in the isotropic liquid phase of 5CB, a nematogenic LC compound with a permanent dipole moment parallel to the long molecular axis. The following behavior was indicated [87]:(10)$${\Delta \varepsilon }^{\prime }\left(f,T\right)=\varepsilon^{\prime }\left(f,T\right)-\varepsilon \left(T\right)=a+bT\;\mathrm{for}\;T\;\to {\;T}_{I–N},$$
where $\varepsilon =\varepsilon^{\prime }\left(f_{stat.},T\right)$ is for the dielectric constant associated with the frequency in the middle of the static domain, and $\varepsilon^{\prime }\left(f,T\right)$ is for frequencies $f<f_{stat.}$ and $f\ll f_{stat.}$. In refs. [31,32,87], the evolution of $\varepsilon \left(T\right)$ was linked to the critical-like behavior given by Equation (2).

The gradual reduction in the linear domain with decreasing frequency in the LF domain was noted. The linear behavior was interpreted as the LF manifestation of pretransitional/prenematic fluctuations. Recently, such behavior was confirmed in the isotropic liquid phase of 8OCB, 11CB, and 5CB [31,32,33,87] and their nanocolloids in pressure-related tests.

The same behavior of ${\Delta \varepsilon }^{\prime }\left(f,T\right)$ occurs for MBBA with the transverse permanent dipole moment and its nanocolloids, as shown in Figure 10. However, the above image is only an apparent picture. When applying a semi-log scale, reducing the factor that changes ${\Delta \varepsilon }^{\prime }(f,T)$ covers several decades, and the following description emerges in Figure 11:(11)$${\Delta \varepsilon }^{\prime }\left(f,T\right)=\Delta_{0}^{f}exp\left(F\times T\right),$$
where $\Delta_{0}^{f}$ means the constant prefactor, associated with the given frequency f and F=const.

Weak distortions, diminishing with the addition of nanoparticles, from this behavior are visible for frequencies from 468 Hz to 5 kHz. Notably, such a presentation reveals LF processes that impact even the frequency f=53 kHz, often considered ‘purely static’.

Notably, the portrayal associated with Equation (11) obeys in the isotropic, even up to $\sim T_{I–N}+70$. For High-concentration nanocolloids and low-frequencies, it appears in the nematic phase, even down to the N–S transition. For low-concentration nanocolloids, particularly pure MBBA, and the highest tested frequencies, the behavior described by Equation (11) appears in the nematic phase but with the opposite sign of the amplitude F<0.

### 3.5. Dynamic Properties in MBBA and Related Nanocolloids

DC electric conductivity is a metric of dynamic properties, mainly associated with translational processes [72,74]. It also reflects dynamics associated with orientational processes, as shown by Debye–Stokes–Einstein (DSE) $\;\sigma \left(T\right)\tau \left(T\right)=const.$ or fractional DSE (fDSE) $\sigma \left(T\right){\left[\tau \left(T\right)\right]}^{\varphi }=const.$ laws, confirmed experimentally in LC materials, including nanocolloids [30,32]. For experimental data considered in the given report, the significant advantage of DC electric conductivity is its unequivocal determination from $\varepsilon^{\prime \prime }\left(f\right)$ spectra in the isotropic liquid and nematic phases (see Appendix A).

Figure 12 shows the temperature dependence of MBBA and its nanocolloids with BaTiO_3_ nanoparticles in the Arrhenius scale ${{log}_{10}\sigma }^{-1}$ vs. 1/T, where the basic Arrhenius dependence, with the activation energy $E_{\sigma }=const.$, is associated with the linear behavior. The addition of nanoparticles increases the electric conductivity (decreases $\sigma^{-1}$), which is particularly visible for x=0.5% and x=1% concentrations. Generally, the rise and decrease in electric conductivity are evidenced in different LC-based nanocolloids [8,9,10,97,98]. The authors of this report noted that for nCB doped with BaTiO_3_ nanoparticles, both patterns, depending on the concentration of nanoparticles, appeared [31]. Explanations of the phenomenon recall the most common definition of DC electric conductivity as the ability to transport direct electric current, depending on the number of free electrons or ionic species within the material and their mobility. In liquid crystalline systems, they are heuristically called ‘residual ionic contaminations’ and are linked to the post-manufacturing remaining or consequences of material/s degradation [8,9,10,97,98]. This means that they are not precisely defined and differ from the basic LC molecules. Consequently, it is stated that some nanoparticles can ‘supplement’ or ‘trap’ residual ions to explain the above behavior [8,97,98]. Is such an explanation, essentially general and heuristic, in agreement with the basic experimental evidence? In the opinion of the authors, the answer is not clear.

The DC electric conductivity can be determined from the low-frequency part of the imaginary part of dielectric permittivity as $\sigma =\omega {\varepsilon_{0}\varepsilon^{\prime \prime }}^{\;}\left(f\right)=2\pi f\varepsilon_{0}\varepsilon^{\prime \prime }\left(f\right)=const.$ [72], and manifests as the horizontal line in the $\sigma^{\prime }\left(f\right)$ spectrum (see the Appendix and ref. [31]). It is directly coupled to the primary relaxation time τ, determined from the peak of the loss curve in the high-frequency part of the $\varepsilon^{\prime \prime }\left(f\right)$ spectrum via the Debye–Stokes–Einstein (DSE) law: $\sigma \left(T\right)\tau \left(T\right)=const$. In complex systems where the broadening of the loss curve above the basic Debye pattern takes place, the fractional DSE law appears to be $\sigma \left(T\right){\left[\tau \left(T\right)\right]}^{\varphi }=const.$ [30]. It can occur in systems that tend toward multimolecular aggregation, like pretransitional fluctuations [30,31,32,33]. Such a direct link between orientations-related primary relaxation time and transport (shift)-related DC electric conductivity can be explained only if they are associated with the same element—in the given case, the rotation (τ) and shift (σ) of LC molecule with respect to the equilibrium position. In such frames, post-manufacturing ‘residual ionic contaminations’ species different from LC molecules should manifest as the violation from the DC conductivity-related ‘horizontal’ behavior mentioned above and observed in experiments [31]. Taking this picture into account and the significance of pretransitional effects coupled to multimolecular fluctuation, one can consider the fundamental discussion regarding DC conductivity changes in LC-based nanocolloids focused on the interaction between given nanoparticles, taking into account the type and concentrations, with critical-like fluctuations.

Figure 12 shows such temperature dependence for MBBA and its nanocolloids with BaTiO_3_ nanoparticles in the Arrhenius scale ${{log}_{10}\sigma }^{-1}$ vs. 1/T where the basic Arrhenius dependence, with the activation energy $E_{\sigma }=const$, manifests via a linear behavior. The behavior visible in Figure 12 is explicitly non-linear, suggesting the super-Arrhenius (SA) pattern with the temperature-dependent activation energy. It is most often parameterized by the Vogel–Fucher–Tammann (VFT) dependence, namely [99,100,101,102]:(12)$$\sigma^{-1}\left(T\right)=\sigma_{0}exp\left(\frac{E_{\sigma }\left(T\right)}{RT}\right)\;\;\;\Rightarrow \;\;\;\;\sigma^{-1}\left(T\right)=\sigma_{0}exp\left(\frac{A_{VFT}}{T-T_{0}}\right),$$
where the left part is for the general SA dependence and the right part is for its VFT replacement equation; $A_{VFT}=const$, and $T_{0}$ denotes the extrapolated singular temperature in glass-forming liquid often associated with the so-called Kauzmann temperature and located below the glass temperature.

The functional ‘flexibility’ of the VFT dependence allows for an effective parameterization of the results presented in Figure 12. However, the results presented in Figure 13 show that VFT enables only effective parameterization, not justified for a given set of experimental data.

Figure 13 presents the results of the distortions-sensitive analysis focused on the temperature dependence of the apparent activation enthalpy $H_{a}\left(T\right)$ [100], which can be alternatively considered as the steepness index showing relative changes in $\sigma^{-1}\left(T\right)$ for the Arrhenius scale presentation [102]:(13)$${H_{a}\left(T\right)=\left[\frac{dln\sigma^{-1}\left(T\right)}{d\left(1/T\right)}\right]}^{-1}=HT+HT^{+}\;\Rightarrow \;\frac{dln\sigma^{-1}\left(T\right)}{d\left(1/T\right)}=\frac{H}{T-T^{+}},$$
where H=const and $T^{+}$ is the extrapolated singular temperature associated with ${\left[H_{a}\left(T^{+}\right)\right]}^{-1}=0$, and $HT^{+}=const.$ condition.

Such behavior leads directly to the ‘activated-critical’ relation recently introduced by one of the authors (ADR) for portraying reference data [102]:(14)$$\sigma^{-1}\left(T\right)=C_{\Gamma }{\left(\frac{T-T^{+}}{T}\right)}^{-\Gamma }{\left[\exp\;\left(\frac{T-T^{+}}{T}\right)\right]}^{\Gamma }=C_{\Gamma }{\left(t^{-1}\exp\;t\right)}^{\Gamma },$$
where $t=\left(T-T^{+}\right)/T$, the prefactor $C_{\Gamma }=const.$, and the exponent Γ=const.

The behavior evidenced in Figure 13 excludes the validated description of $\sigma \left(T\right)$ data shown in Figure 12 using the VFT equation, for which the linear behavior of ${\left[H_{a}\left(T\right)\right]}^{-1/2}$ vs. 1/T plot should occur [102].

One of the properties hardly considered for the fundamental model analysis of liquid crystalline materials, including nanocolloids, is the tangent of the loss angle δ, often recalled as tanδ and defined as follows [72,103,104,105,106,107,108]:(15)$$tan\delta \left(f,T\right)=\frac{\varepsilon^{\prime \prime }\left(f,T\right)}{\varepsilon^{\prime }\left(f,T\right)}=\frac{i_{s}}{I_{0}+\Delta i},$$
where $I_{0}=\omega C_{0}U$ is the current applied to the capacitance with the dielectric from the external source, $C_{0}$ is the capacitance of the empty capacitor, and U is the applied voltage; $\Delta i=i\omega \chi^{\prime }C_{0}U$ is associated with the presence of the dielectric in the capacitor and $i_{s}=\omega \varepsilon^{\prime \prime }C_{0}U$ is the loss current in the dielectric.

Pure MBBA shows strong manifestations of both I–N and N–S transitions, which gradually diminish in nanocolloids when the amount of nanoparticles increases. The frequency f=1 kHz can be considered the terminal of the static domain, with only a minor impact of low-frequency domain effects (see Section 2). Generally, tanδ of a material denotes the dissipation of electrical energy due to different physical processes such as electric conductivity or dielectric relaxation. It is also expressed as the dissipation factor D or the quality factor D=tan δ=1/Q.

This magnitude enables the estimation of the power loss, which can be converted to heat:(16)$$P=Qtan\delta =\omega CV^{2}tan\delta =\varepsilon_{0}\varepsilon^{\prime \prime }E^{2},$$

Notable is also the link to the real and imaginary parts of dielectric permittivity:(17)$$\varepsilon^{*}=\varepsilon^{\prime }-i\varepsilon^{\prime \prime }=\varepsilon^{\prime }\left(1-i\times tan\delta \right)$$

The temperature dependence of the dissipation factor for the type of systems discussed in the given report has been hardly (if at all) considered so far. To the best of the authors’ knowledge, preliminary consideration has only been presented for linseed oil, a natural material showing dielectric properties similar to the isotropic phase of LC materials [65]. Figure 14, Figure 15 and Figure 16 show the $D\left(T\right)$ evolution at three selected frequencies measured in MBBA and MBBA + NP nanocolloids. For the highest frequency, f=1.23 MHz, the relaxation process can be important. The pattern of changes is different for frequencies near the static (f=1 kHz) and LF (f=12 Hz) domains:(18)$$D\left(T\right)=tan\delta =\delta^{*}+d\left(T-T_{\delta }\right)+D{\left(T-T_{\delta }\right)}^{\psi },$$
where $\delta^{*},d,D=const.$ and the exponent ψ≈2.8 for f=1 kHz and ψ≈4 for f=12 Hz.

The behavior described by Equation (18) in Figure 15 and Figure 16 is obeyed in pure MBBA and all tested nanocolloids. The addition of nanoparticles changes only the prefactor $\delta^{*}$. Notable is the smooth passing of the I–N transition without any hallmark.

The authors stress that Equation (18) presents fair but empirical parameterization, whose explanation requires further studies.

## 4. Conclusions

The report discusses static and low-frequency dielectric properties in nematogenic MBBA and related nanocolloids with paraelectric BaTiO_3_ nanoparticles (spherical, *d* = 50 nm). The distortions-sensitive analysis of the dielectric constant revealed the hidden anomaly of the dielectric constant, strongly influenced by the addition of nanoparticles, which finally leads to the ‘anomalous’ pretransitional anomalies for *x* = 0.5% and *x* = 1% concentrations of NPs. The evolution of the dielectric constant in the nematic phase indicates its split into two regions, with the crossover related to the standard ‘equilibrium, hidden melting temperature. Notable is the finding of the exponential behavior of the low-frequency contribution to the real part of the dielectric permittivity, which has been not reported so far. The next issue is the critical-like pretransitional behavior in the solid phase and the strong rise in this effect when adding nanoparticles. For dynamics, the explicit preference for the ‘hyperbolic’ or ‘critical’ evolution of the apparent activation enthalpy is worth stressing, leading to the preference for the ‘activated & critical’ equation introduced recently [102]. Finally, worth stressing is the long-range empirical ‘critical-like’ behavior of the dissipation factor (*D* = *tgδ*).

This report shows that pretransitional behavior, matched to multimolecular critical-like fluctuations, is essential for understanding and describing the behavior of dielectric properties in the isotropic, nematic, and solid phases of MBBA and related nanocolloids. Notably, MBBA is the nematogenic compound with the transverse dipole moment. Hence this report supplements the existing evidence for nanocolloids associated with the ‘parallel’ dipole moment [27,28,29,30,31,32,33]. They explicitly show that when discussing the properties of LC + NP nanocolloids, one should consider the distance from the nearest phase transition or the temperature to properly take into account the impact of critical-like fluctuations. Some of the results presented, particularly associated with the low-frequency domain and the dissipation factor, can be considered the gateway for further experimental and modeling studies.

## Figures and Tables

**Figure 1 nanomaterials-14-00655-f001:**
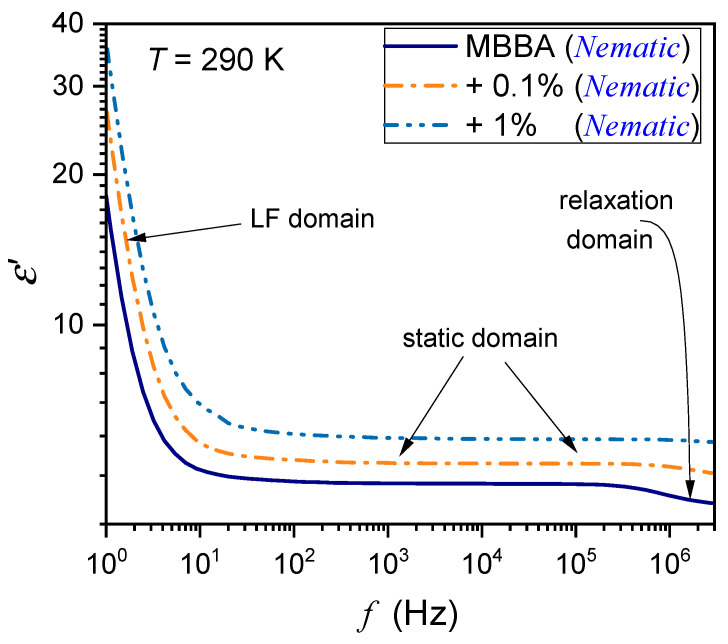
The frequency spectrum of the real part of dielectric permittivity in the nematic phase of MBBA and MBBA + BaTiO_3_ nanocolloids. The behavior associated with characteristic frequency domains is indicated. LF stands for low-frequency, and the static domain is related to the dielectric constant.

**Figure 2 nanomaterials-14-00655-f002:**
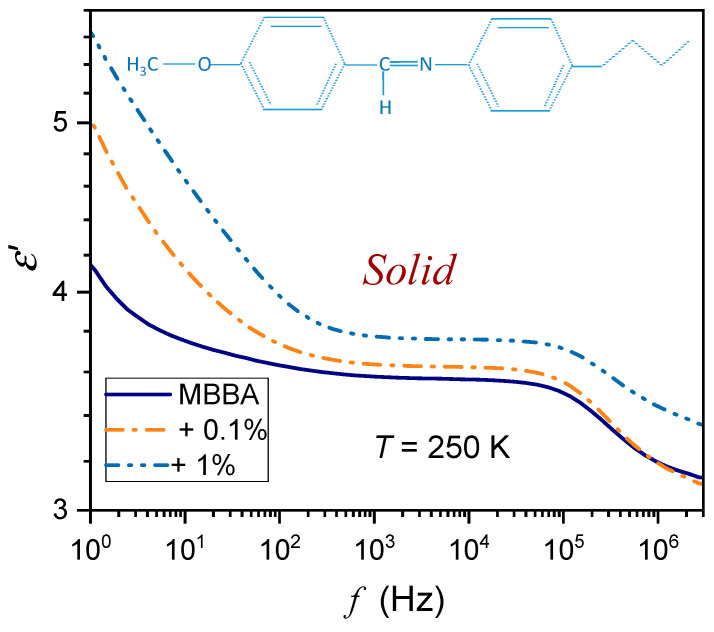
The frequency spectrum of the real part of dielectric permittivity in the solid phase of MBBA and MBBA + BaTiO_3_ nanocolloids, close to the N–S transition. Note the log-log scale. The skeletal formula for MBBA is also shown in the picture.

**Figure 3 nanomaterials-14-00655-f003:**
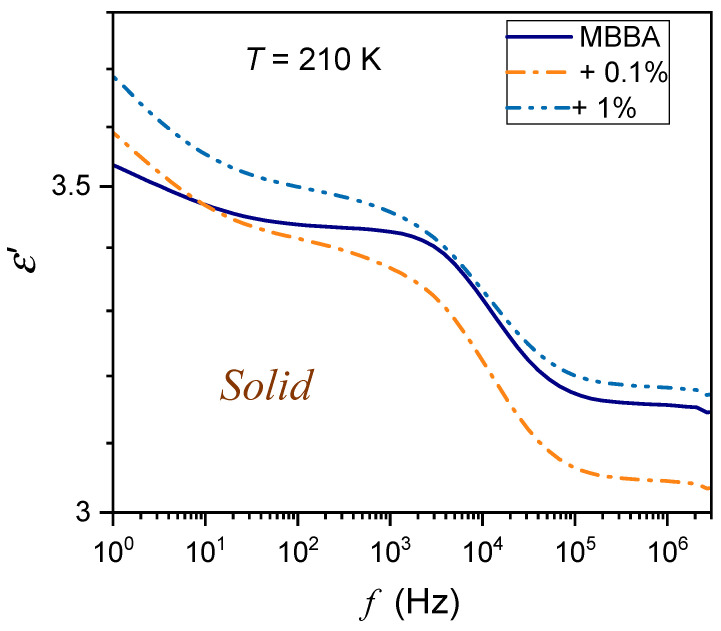
The frequency spectrum of the real part of dielectric permittivity in the solid phase of MBBA and MBBA + BaTiO_3_ nanocolloids, remote from the N–S transition.

**Figure 4 nanomaterials-14-00655-f004:**
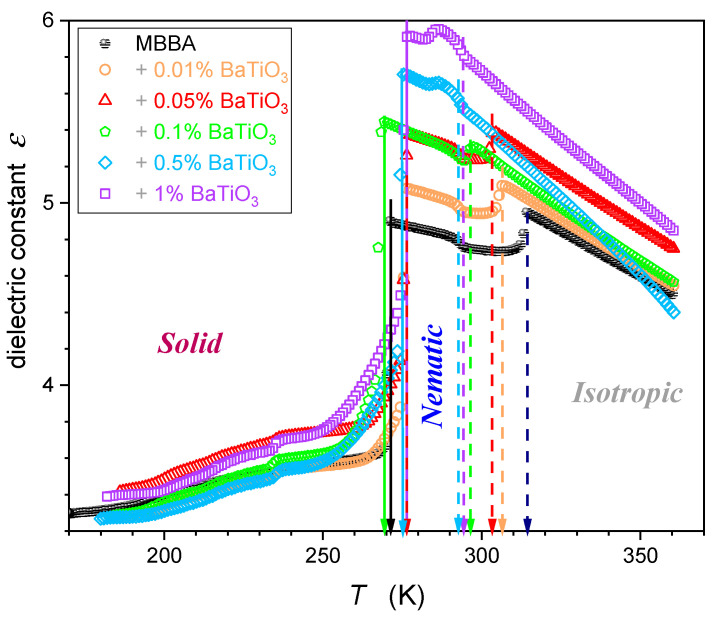
Dielectric constant temperature changes in MBBA and its nanocolloids, for nanoparticle concentrations given in the plot. Dashed arrows indicate transitions between the isotropic liquid and nematic phases, and solid arrows indicate nematic–solid phase transitions.

**Figure 5 nanomaterials-14-00655-f005:**
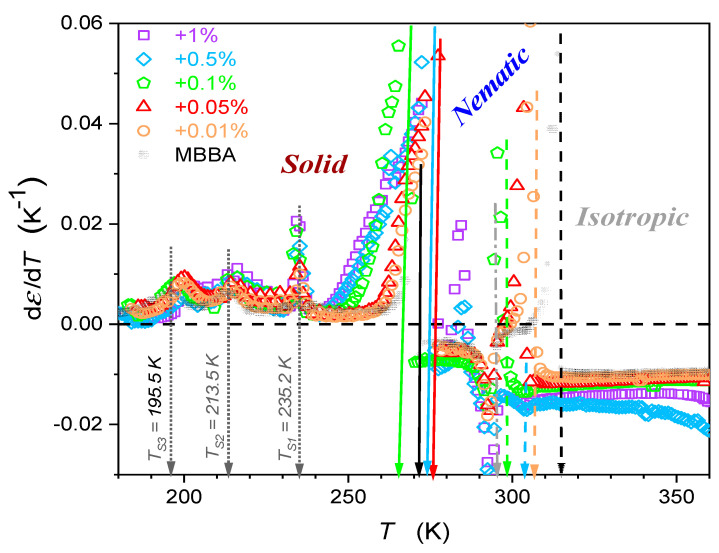
Temperature changes in dielectric constant derivatives of MBBA and its nanocolloids with BaTiO_3_ nanoparticles. Concentrations are given in the plot. Dashed arrows indicate transitions between the isotropic liquid and nematic phases, and solid arrows indicate nematic–solid phase transitions. Values of $T_{I–N}$ and $T_{N–S}$ temperatures are given in Table 1. Dotted arrows indicate phase transition in the solid phase, with $T_{S1},\;\;T_{S2},\;{\;T}_{S3}$ phase transition temperatures. The dashed-dotted arrow (in grey) indicates the transformation within the nematic phase, associated with $T_{cross.}\approx 295.3\;K$.

**Figure 6 nanomaterials-14-00655-f006:**
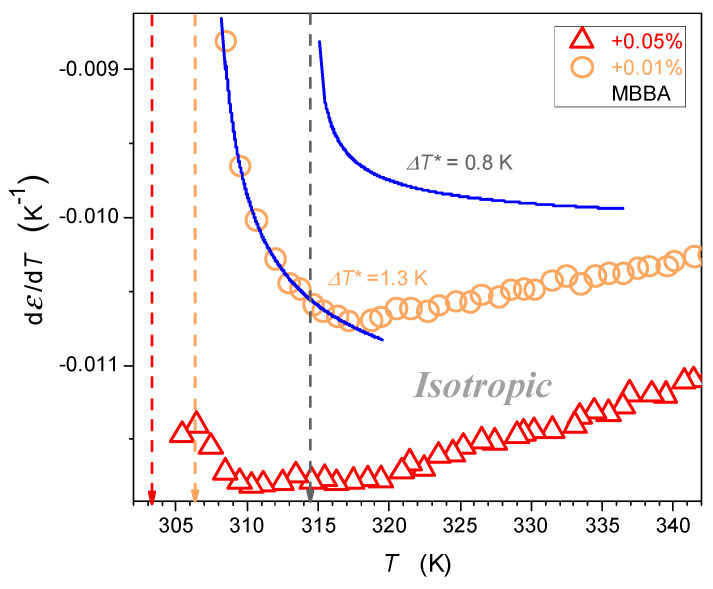
The focused insight on the behavior of dielectric constant in the isotropic liquid phase in MBBA and related nanocolloids for small concentrations of nanoparticles. Solid curves are associated with the parameterization via Equation (5). Reference data are taken from Figure 4. The dashed arrows indicate the clearing temperatures, their colors correspond to the colors of the data series.

**Figure 7 nanomaterials-14-00655-f007:**
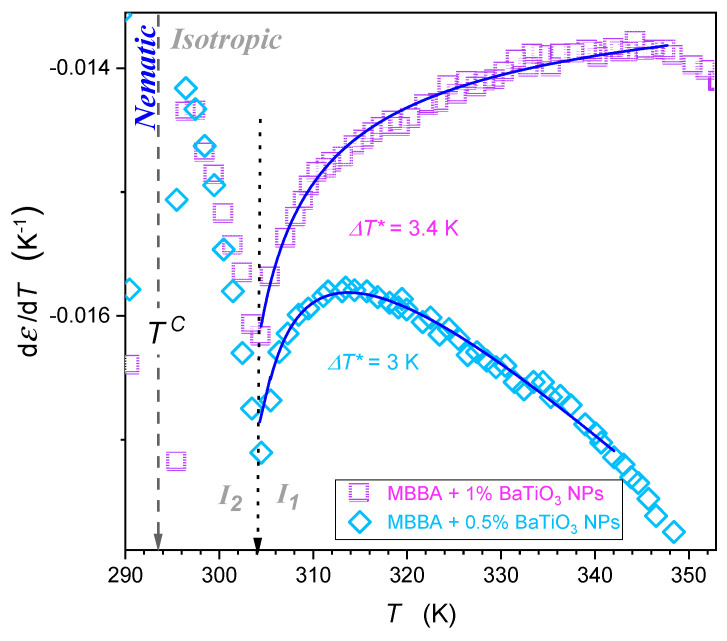
The focused insight for the behavior of dielectric constant in the isotropic liquid phase in MBBA-based nanocolloids for larger concentrations of nanoparticles. Solid curves are associated with the parameterization via Equation (5). Reference data are taken from Figure 4. The gray dashed arrow indicates the isotropic-nematic phase transition temperature $T^{C}$. Note: the transition occurs within the isotropic liquid phase, which suggests the liquid-liquid transition ($I_{2}\leftarrow I_{1}$)—indicated by the dotted arrow.

**Figure 8 nanomaterials-14-00655-f008:**
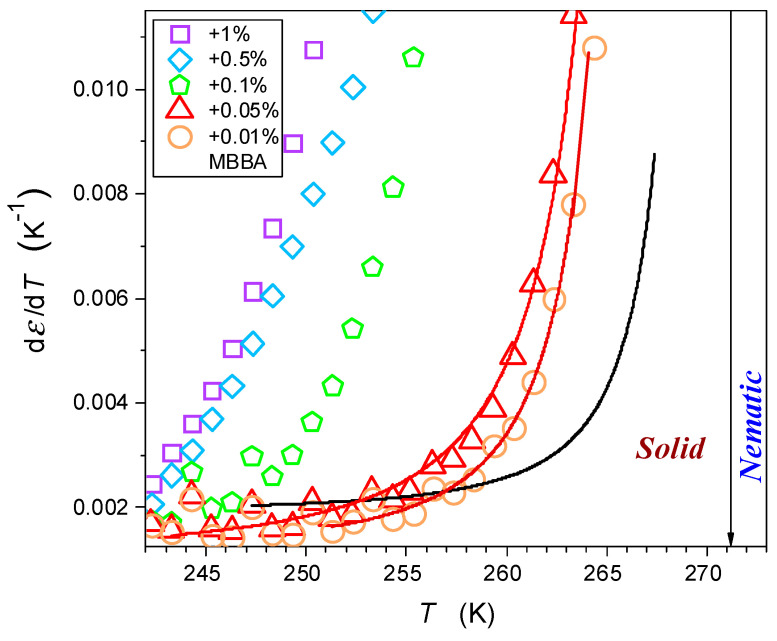
The pretransitional changes in dielectric constant in the solid phase of MBBA and its nanocolloids with BaTiO_3_ nanoparticles. Solid curves are associated with Equation (8), with singular temperature $T_{m.f}^{*}$ in the liquid phase at $\sim T_{m,f}+0.5$. The arrow indicate the nematic-solid phase transition temperature.

**Figure 9 nanomaterials-14-00655-f009:**
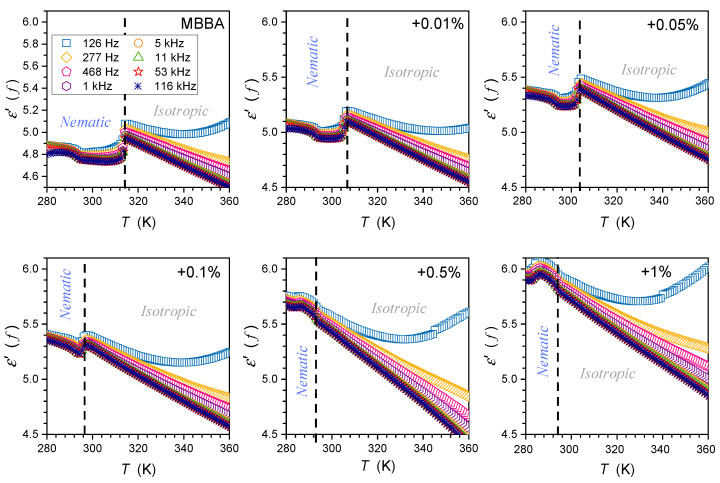
Temperature changes in the real part of dielectric permittivity in the isotropic liquid and nematic phases of MBBA and its nanocolloids with BaTiO_3_ nanoparticles for concentration (in mass fractions) frequencies given in the plot. The latter extends from the static to the low-frequency domain. For each figure, the same ranges are presented.

**Figure 10 nanomaterials-14-00655-f010:**
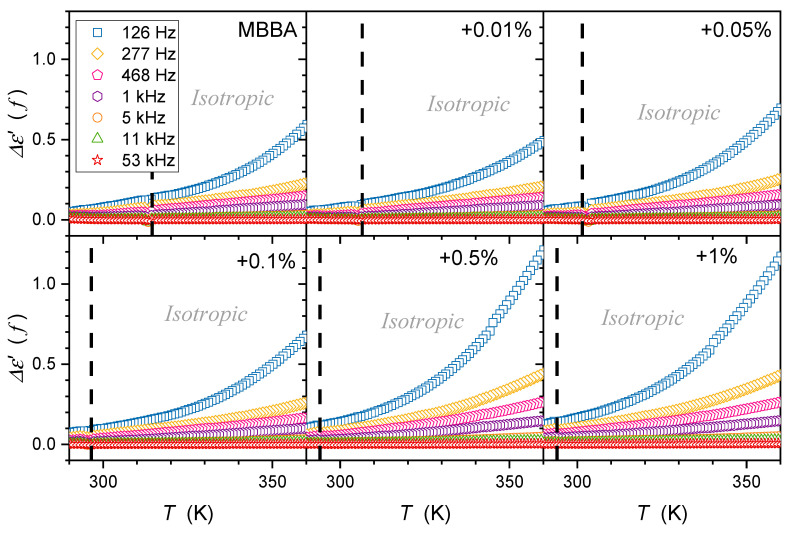
Temperature evolutions of the difference ${\Delta \varepsilon }^{\prime }=\varepsilon^{\prime }\left(f\right)-\varepsilon$, where $\varepsilon =\varepsilon^{\prime }\left(f=116\;\mathrm{kHz}\right)$ is for dielectric constant—related to the static domain. Note the link to Equation (10). Dashed lines show the ‘average’ position of the clearing temperature for all concentrations.

**Figure 11 nanomaterials-14-00655-f011:**
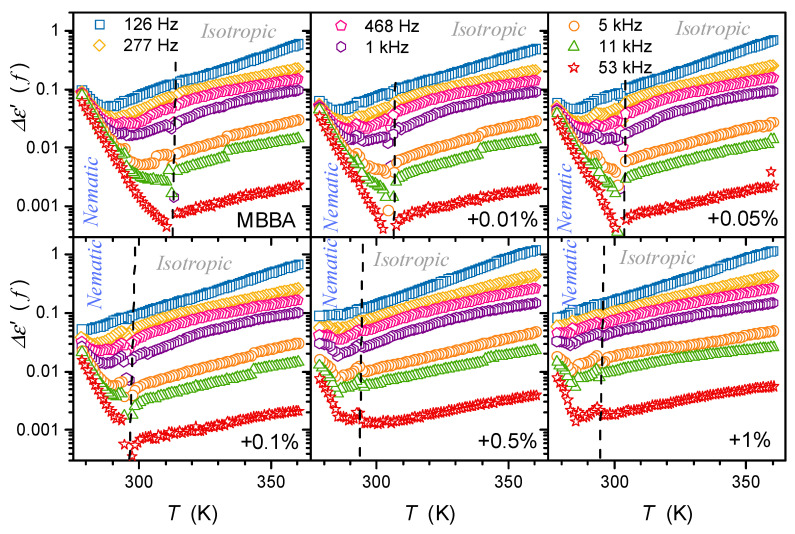
Results from Figure 10, presented as ${\Delta \varepsilon }^{\prime }\left(T\right)=\varepsilon^{\prime }\left(f\right)-\varepsilon$ dependencies for MBBA and its nanocolloids with BaTiO_3_ in the semi-log scale. The plots reveal the hidden exponential behavior (Equation (11)) of ${\Delta \varepsilon }^{\prime }\left(T\right)$ in the low-frequency domain. Dashed lines show the ‘average’ position of the clearing temperature for all concentrations.

**Figure 12 nanomaterials-14-00655-f012:**
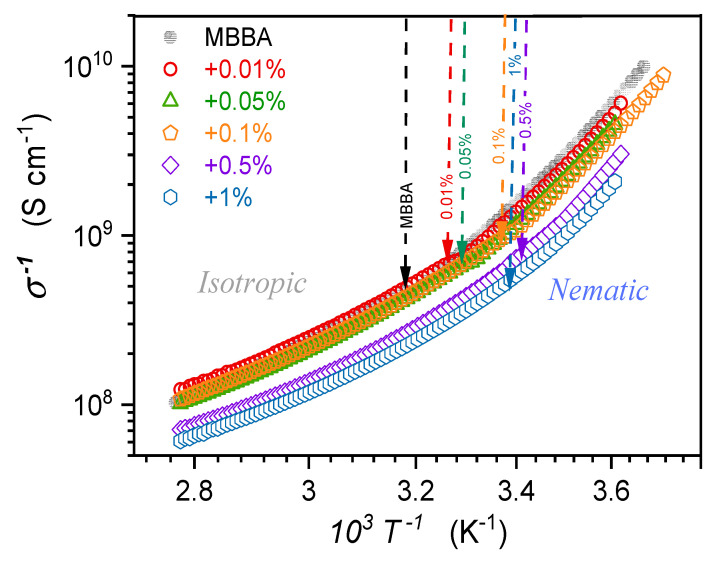
The temperature evolution of DC electric conductivity in the isotropic liquid and nematic phases of MBBA and nanocolloids with BaTiO_3_ nanoparticles. Arrows indicate clearing temperatures for pure MBBA and related nanocolloids, with given concentrations of nanoparticles.

**Figure 13 nanomaterials-14-00655-f013:**
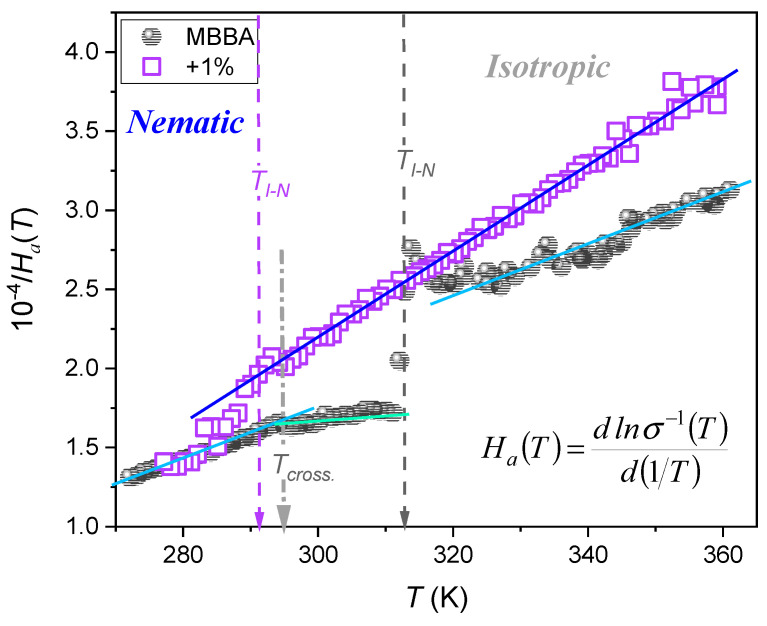
The temperature evolution of the reciprocal of the apparent enthalpy (definition in the plot), and alternatively, the steepness index for DC electric conductivity in MBBA and related nanocolloids with 0.1% BaTiO_3_ nanoparticles. Note the link to Equation (13). The plot recalls the analysis introduced in ref. [102] for ‘glassy’ dynamics.

**Figure 14 nanomaterials-14-00655-f014:**
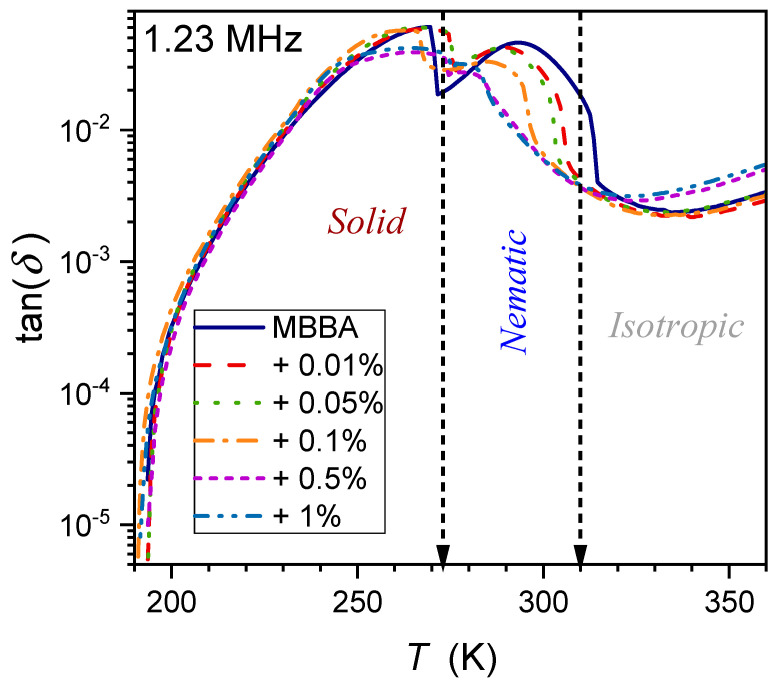
Temperature evolutions for tgδ loss factor (frequency f=1.23 MHz) in MBBA and related nanocolloids with BaTiO_3_ nanoparticles.

**Figure 15 nanomaterials-14-00655-f015:**
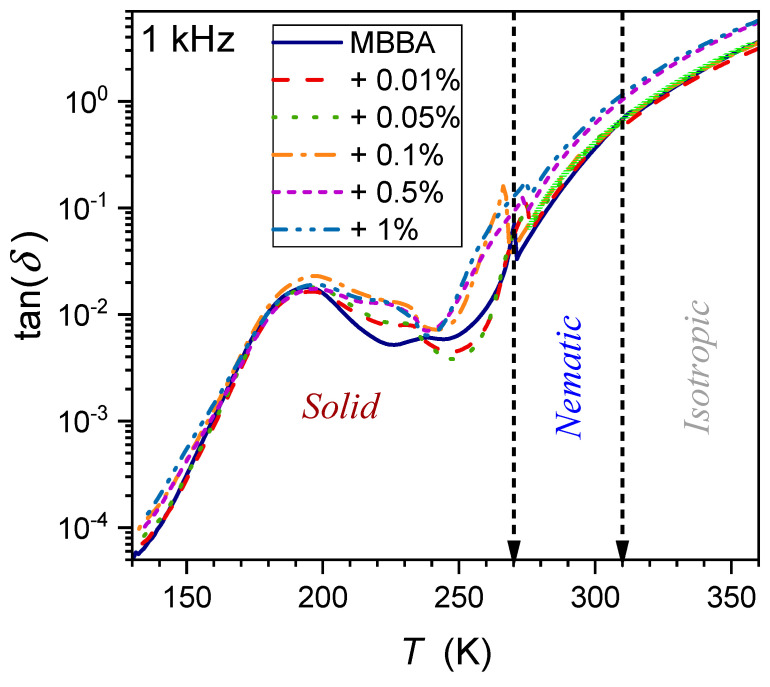
Temperature evolutions for tgδ loss factor (frequency f=1 kHz) in MBBA and related nanocolloids with BaTiO_3_ nanoparticles. The solid green curve is related to Equation (18) with the exponent ϕ≈2.8.

**Figure 16 nanomaterials-14-00655-f016:**
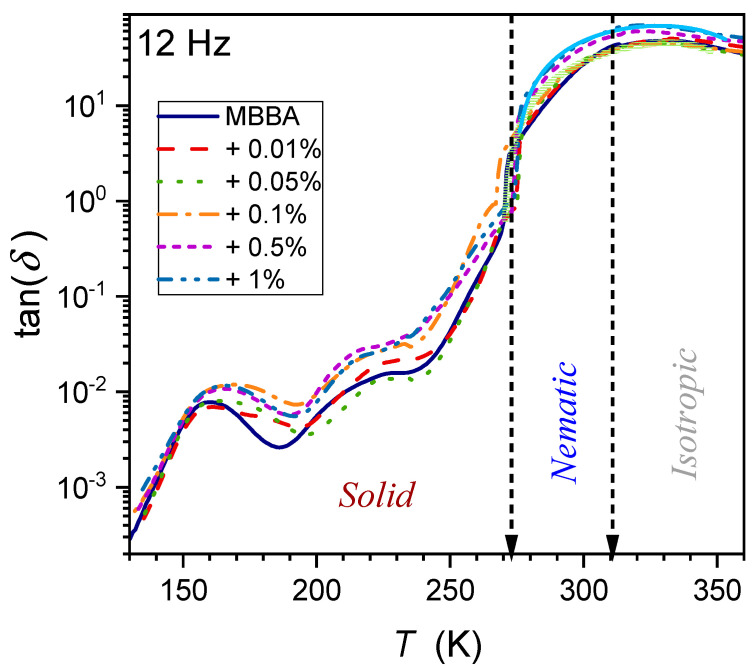
Temperature evolutions for tgδ loss factor (frequency f=12 Hz) in MBBA and related nanocolloids with BaTiO_3_ nanoparticles. The solid green curve is related to Equation (18) with the exponent ϕ≈4.

**Table 1 nanomaterials-14-00655-t001:** Basic phase transition temperatures in MBBA and its nanocolloids with BaTiO_3_ nanoparticles. $T_{I–N}$ is for the weakly discontinuous Isotropic–Nematic (I–N) transition temperature (clearing temperature) and $T_{N–S}$ is for the strongly discontinuous Nematic–Solid (N–S) transition temperature. Concentrations are given in mass fraction: $x\%=\left[m_{NPs}/\left(m_{NPs}+m_{MBBA}\right)\right]\times 100\%$ and the volume fraction: *ϕ*% $=\left[V_{NPs}/\left(V_{NPs}+V_{MBBA}\right)\right]\times 100\%$. The latter is associated with pure MBBA clearing temperature.

System: MBBA + NPs	$T_{I–N}(K)$	$T_{N–S}(K)$
MBBA (*x*%, *ϕ*% = 0)	314.4	271.2
+0.01% (*x*%); +0.17 (*ϕ*%)	306.3	276.5
+0.05% (*x*%); +0.008% (*ϕ*%)	303.3	276.1
+0.1% (*x*%); +0.017% (*ϕ*%)	296.6	269.1
+0.5% (*x*%): +0.088% (*ϕ*%)	292.9	275.1
+1%(*x*%): +0.17% (*ϕ*%)	293.7	276.6

**Table 2 nanomaterials-14-00655-t002:** Values of parameters portraying $d\varepsilon \left(T\right)/dT$ pretransitional changes in the isotropic phase of MBBA and related nanocolloids with BaTiO_3_ nanoparticles via Equation (5). Graphical results are shown by blue curves in Figure 6 and Figure 7.

**System**	a	C	${\Delta T}^{*}(K)$	*α*
MBBA	−0.0101	9.14 × 10^−4^	0.8	1/2
+0.01%	−0.0117	3.13 × 10^−3^	1.3	1/2
+0.5%	−0.011	−6.91 × 10^−3^	3.0	1/2
+1%	−0.013	−6.12 × 10^−3^	3.4	1/2

## Data Availability

All data are available directly from the authors following reasonable request. They are also deposited in the public open-access REPOD database supporting data storage for research carried out at the Institute of High-Pressure Physics of the Polish Academy of Sciences.

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
