# Peer review of "The Dominance of Pretransitional Effects in Liquid Crystal-Based Nanocolloids: Nematogenic 4-methoxybenzylidene-4′–butylaniline with Transverse Permanent Dipole Moment and BaTiO3 Nanoparticles"

_nanomaterials, 2024, doi:10.3390/nano14080655_

Round 1

Reviewer 1 Report

Comments and Suggestions for Authors

In this manuscript, the author attempted to introduce the dielectric properties of liquid crystal molecule 4-methoxybenzyl4 '- butylphenylamine (MBBA) with transverse permanent dipole moment and MBBA mixture doped with a small amount of nanoferroelectric barium titanate (spherical, d=50 nm). The static, low-frequency, and dynamic dielectric properties of MBBA before and after doping were studied in detail in isotropic, nematic, and solid-state states. It is interesting that the dielectric constant of MBBA before and after doping is strongly influenced by frequency, temperature, and nanoparticle content. I recommend publishing this manuscript, however, before it is published, it should be well revised. The following are some issues that need to be noted:
1.For the convenience of readers, the author should provide the molecular structure of MBBA
2. It is recommended that the author use a universal three line format to modify Tables 1 and 2
3.The two horizontal axes in Figure 12 are confusing and difficult to understand
4 All references should be consistent with the format of the journal

Comments on the Quality of English Language

In this manuscript, the author attempted to introduce the dielectric properties of liquid crystal molecule 4-methoxybenzyl4 '- butylphenylamine (MBBA) with transverse permanent dipole moment and MBBA mixture doped with a small amount of nanoferroelectric barium titanate (spherical, d=50 nm). The static, low-frequency, and dynamic dielectric properties of MBBA before and after doping were studied in detail in isotropic, nematic, and solid-state states. It is interesting that the dielectric constant of MBBA before and after doping is strongly influenced by frequency, temperature, and nanoparticle content. I recommend publishing this manuscript, however, before it is published, it should be well revised. The following are some issues that need to be noted:
1.For the convenience of readers, the author should provide the molecular structure of MBBA
2. It is recommended that the author use a universal three line format to modify Tables 1 and 2
3.The two horizontal axes in Figure 12 are confusing and difficult to understand
4 All references should be consistent with the format of the journal

Author Response

  1. Reviewer:For the convenience of readers, the author should provide the molecular structure of MBBA’

Response:  The structure is now shown in Figure 2

  1. Reviewer: ‘It is recommended that the author use a universal three line format to modify Tables 1 and 2

Response:  Tabs. I and II have been corrected/re-formatted.

  1. Reviewer:The two horizontal axes in Figure 12 are confusing and difficult to understand’

Response: Now, there is only a single horizontal axe. Moreover, arrows indicating clearing temperatures and names of phases were introduced to clarify further the picture.

  1. Reviewer: ‘All references should be consistent with the format of the journal’.

Response: All references have been corrected to meet Nanomaterials pattern.

Additionally, the paper has been deeply cleaned to avoid misprints and to improve grammar.

Reviewer 2 Report

Comments and Suggestions for Authors

The paper under review reports the results of extensive experimental studies of pretransitional effects in nematic liquid crystal MBBA (both pure and doped with barium titanate paraelectric nanoparticles). The authors are well-known experts in this subfield of liquid crystal research and they build on their previous work. As a general statement, the paper assuming some revision would be of interest to liquid crystal scientists, and I support its publication.

Comments to address:

1)      Given the fact that liquid crystals are highly anisotropic materials, the authors should provide some information related to the quality of alignment of the studied samples. The utilized cell thickness (0.2 mm) implies that the samples were not aligned. If this is the case, the measured effective values of dielectric constants of undoped and doped liquid crystals can be affected by both different qualities of alignment and by the presence of nanoparticles. At least, some reasonable discussion of this issue should be provided in the revised paper.

2)      The discussion of ionic impurities in the studied samples is very brief.  Figure 12 indicates that barium titanate nanoparticles result in the increase in the DC electrical conductivity. Similar effects were reported in many papers, and the authors could comment on this issue a bit more. Figure 12 suggest the ion releasing regime in the case of MBBA doped with barium titanate nanoparticles. The authors could discuss this effect considering existing literature and comment on possible sources of the ionic contamination or ion leaching in barium titanate nanoparticles.

3)      It is not clear whether equation (18) empirical or has some theoretical justification.

4)      Double check all figures. For example, Figure 1 shows data collected at 290 K. This temperature corresponds to a crystalline state however Figure 1 says it is nematic state.

Author Response

1) Reviewer #2: Given the fact that liquid crystals are highly anisotropic materials, the authors should provide some information related to the quality of alignment of the studied samples. The utilized cell thickness (0.2 mm) implies that the samples were not aligned. If this is the case, the measured effective values of dielectric constants of undoped and doped liquid crystals can be affected by both different qualities of alignment and by the presence of nanoparticles. At least, some reasonable discussion of this issue should be provided in the revised paper.

Response: These issues are commented on and explicitly explained at the beginning of the Results and Discussion section, lines 192-212

2)      Reviewer #2:  The discussion of ionic impurities in the studied samples is very brief. Figure 12 indicates that barium titanate nanoparticles result in the increase in the DC electrical conductivity. Similar effects were reported in many papers, and the authors could comment on this issue a bit more. Figure 12 suggest the ion releasing regime in the case of MBBA doped with barium titanate nanoparticles. The authors could discuss this effect considering existing literature and comment on possible sources of the ionic contamination or ion leaching in barium titanate nanoparticles.

Response: This is one of the basic problems of LC-based nanocolloids, and the proposed explanation is rather heuristic in the opinion of the authors. Recalling them, we also propose a new picture which in our opinion better fits some significant, but most often omitted experimental facts. It is presented in the "3.4 Dielectric permittivity in the low-frequency domain" section, between lines 474-510.

3)   Reviewer #2: It is not clear whether equation (18) empirical or has some theoretical justification.

Response: It is the first (ever) evidence focused on the Dissipation factor in such a way. Eq. (18) surprisingly well portrays experimental data in the isotropic and nematic phases. But now, this is an empirical finding waiting for further studies and theory. It is explicitly stated at the end of the Results and Discussion section, just above the Conclusions.

4)      Reviewer #2:  Double check all figures. For example, Figure 1 shows data collected at 290 K. This temperature corresponds to a crystalline state however Figure 1 says it is nematic state.

Response:  It has been checked and corrected. Please note that measured nematic-solid phase transition temperatures are lower than the standard N-S melting temperature (see Table I).

Reviewer 3 Report

Comments and Suggestions for Authors

This a very interesting, properly conducted and well-documented study on nanoparticles dispersed in LC matrix. It will certainly deserve publication in Nanomaterials after consideration of following points and completion of the manuscript with some missing information/comments.

 1. In section 2, authors detail the preparation of the dispersions and claim that sedimentation occurs with NP concentrations larger than 1%. When does this sedimentation occur? From my experience, dispersions of large size NPs are not stable down in the isotropic phase to very low concentration, and sedimentation fatally occurs with time when the sonication is stopped. On the contrary, dispersions are often quite stable or metastable in the liquid crystal phase. So, please specify to what refers this 1% concentration limit: sedimentation observed in which phase and after which ageing time of the dispersion. If available, authors should show dispersion stability data versus concentration as it would help to understand the system.

 2. Please, indicate also the NP volume fractions in addition to the NP weight fractions. This gives an idea of the NP volume incorporated at the boundary of domains, which influences the dispersion stability and the NP-matrix effects.

3. In table 2, authors showed the variations of Isotropic-Nematic and Nematic-Solid transitions. What are author's thoughts/comments about this phenomenon? Influence of NP' content on Isotropic-Nematic transition temperature is reported in literature (see for instance Gharbi et al., ACS Applied Nano Materials, 2021, 4 (7), pp.6700).

 4. Many studies report that NPs segregate at the defect lines of liquid crystal matrices (see for instance Jeridi et al., Soft Matter, 2022, 19, 4792). If this segregation occurred here as well, it would impact the behavior of the system. Are author's findings compatible with this segregation? Could it explain the observed dominance of pretransitional effects?

Comments on the Quality of English Language

Quality of language is good, except some spelling errors that could be easily corrected (examples: isotropis, nanocollois).

Author Response

Comments of the Reviewer are related to the central problem of LC-based nanocolloids experimental studies, which are essential for applications. In the opinion of the authors, it remains open and can be addressed in almost any report in this field despite its grand number and decades of studies.

Consequently, the responses below are related only to the type / size / phase of nanoparticles tested in the given report, also based on experiences presented in the recent authors’ report in other (but formally similar) LC  matrixes.

  1. Reviewer #3:In section 2, authors detail the preparation of the dispersions and claim that sedimentation occurs with NP concentrations larger than 1%. When does this sedimentation occur? From my experience, dispersions of large size NPs are not stable down in the isotropic phase to very low concentration, and sedimentation fatally occurs with time when the sonication is stopped. On the contrary, dispersions are often quite stable or metastable in the liquid crystal phase. So, please specify to what refers this 1% concentration limit: sedimentation observed in which phase and after which ageing time of the dispersion. If available, authors should show dispersion stability data versus concentration as it would help to understand the system.

Response:  The issue has been precisely commented on and explained in the Methods section: the text between lines 130 – 159 in the revised manuscript.

  1. Reviewer #3: ‘Please, indicate also the NP volume fractions in addition to the NP weight fractions. This gives an idea of the NP volume incorporated at the boundary of domains, which influences the dispersion stability and the NP-matrix effects.’ 

Response:  In Table I concentrations are given in mass and volume fractions now.

  1. Reviewer #3:In table 2, authors showed the variations of Isotropic-Nematic and Nematic-Solid transitions. What are author's thoughts/comments about this phenomenon? Influence of NP' content on Isotropic-Nematic transition temperature is reported in literature (see for instance Gharbi et al., ACS Applied Nano Materials, 2021, 4 (7), pp.6700).’

Response:  Table I is related to phase transition temperatures, and Table II shows parameters describing the pretransitional effect. The reviewer's comment seems to be related to Table I. The impact of nanoparticles is commented on between lines 130 – 159, where the reference indicated by the Reviewer by Gharbi et al. is also introduced. It is ref. [54] in the revised manuscript.

To reduce the impact of the new, added reference – ref. [57] in the former (basic) version of the manuscript has been removed.

Note also comments in the Introduction section, which indicated the lack of the impact of MPs (the same type and concentrations as in the given report) for n-cyanobiphenyls (nCB) as the LC matrix (‘parallel’ position of the permanent dipole moment). Between lines 238-251 comment/explanation of the situation observed in ‘parallel’ and ‘transverse’ LC materials as the nanocolloids base is presented. These issues were present in the basic version of the report. Regarding the case on the nematic–solid transition, it is the case of the grand cognitive challenge of the general melting/ freezing transition issues, which requires a lot of further studies. Notwithstanding, this report introduces essential new evidence here, as described in Section 3.3.

  1. Reviewer #3: Many studies report that NPs segregate at the defect lines of liquid crystal matrices (see for instance Jeridi et al., Soft Matter, 2022, 19, 4792). If this segregation occurred here as well, it would impact the behavior of the system. Are author's findings compatible with this segregation? Could it explain the observed dominance of pretransitional effects?’

The response:  in the author's opinion, this issue cannot explain the dominance of pretransitional effects. This dominance is well known for ‘pure’ LC compounds, as explicitly presented in the introduction, with extensive references.  Nanoparticles play the role of an ‘endogenic impact’ which can ‘only’ modify the influence of pretransitional fluctuations. Note that pretransitional effects present the experimental evidence showing the impact of multimolecular fluctuations. The microscopic observations, which supplement dielectric studies in nCB presented in refs. [32, 33]   (recalled in the Methods section) does not show such segregation for nanocolloids composed of the given type and concentration of nanoparticles. Hence, in our opinion, this comment goes beyond the given report. It cannot explain the dominance of pretransitional effects, which is clearly related to the basics of the Critical Phenomena Physics: see the classic monography by Anisimov  (ref. [2]), for instance.

  1. Reviewer #3: Quality of language is good, except some spelling errors that could be easily corrected (examples: isotropis, nanocollois).

Response:  It has been corrected. The supplementary ‘text cleaning’ has been carried out.

Round 2

Reviewer 3 Report

Comments and Suggestions for Authors

Authors made all necessary revisions. The manuscript can be accepted in present form.